# Nutritional Composition of Hass Avocado Pulp

**DOI:** 10.3390/foods12132516

**Published:** 2023-06-28

**Authors:** Nikki A. Ford, Paul Spagnuolo, Jana Kraft, Ella Bauer

**Affiliations:** 1Avocado Nutrition Center, 25212 Marguerite Pkwy Ste. 250, Mission Viejo, CA 92692, USA; anc@hassavocadoboard.com; 2Department of Food Science, University of Guelph, 50 Stone Rd., Guelph, ON N1G2W1, Canada; paul.spagnuolo@uoguelph.ca; 3Department of Animal and Veterinary Sciences, The University of Vermont, 570 Main Street, Burlington, VT 05405, USA; jana.kraft@uvm.edu

**Keywords:** *Persea americana*, avocado, macronutrients, vitamins, minerals, fatty alcohols, polyphenols, carotenoids, phytosterols, precision nutrition

## Abstract

Avocados (*Persea americana*) are a unique fruit that can provide health benefits when included in a healthy diet. As health care moves towards precision health and targeted therapies or preventative medicine, it is critical to understand foods and their dietary components. The nutritional composition and plant physiology of the Hass avocado is strikingly different from other fruits. This paper reviews the nutrient and bioactive composition of the edible portion of the Hass avocado (pulp) reported in the literature and from commercial lab analyses of the current market supply of fresh Hass avocados. These results provide comprehensive data on what nutrients and bioactives are in avocado and the quantity of these nutrients. We discuss the reasons for nutrient composition variations and review some potential health benefits of bioactive compounds found in Hass avocados.

## 1. Introduction

Avocados (*Persea americana*) are botanically fruits and consist of a single large seed surrounded by a creamy, smooth-textured, edible fruit (known as pulp or mesocarp) covered by a thick, bumpy skin. The Hass avocado is the most consumed avocado variety in the United States and worldwide [1]. Avocados have a unique physiology and nutritional composition when compared with other fruits. The fruits are picked when mature, but unripe, and the ripening process only begins after removal from the tree [1]. Fresh avocados are eaten when ripe, and often consumed in spreads and dips such as guacamole, alone, as well as a topping for salads, soups, sandwiches, and burgers. The ripe fruit is rich in oleic acid, fiber, micronutrients (e.g., folate, vitamin K, copper, pantothenic acid), and various important phytochemicals, such as lutein, zeaxanthin, and phytosterols [2].

The nutrients and bioactive molecules found in avocados contribute to several health benefits. Data from clinical trials and observational studies have linked avocado consumption with improvements in cardiovascular health, weight control, cognitive function, digestive physiology, and skin health [3,4,5,6]. While these studies are reviewed in detail elsewhere, some key findings are summarized here briefly. Avocado intake improves lipid profiles in adults with dyslipidemia [3] and supports weight management by reducing hunger and increasing meal satisfaction and satiety [3]. Observational studies and clinical trials have found that avocado intake modestly improves cognitive function, especially in frontal cortex executive function [5,7,8]. Avocado intake promotes changes in microbiota composition and fecal metabolites that correlate with a favorable metabolic phenotype in adults with overweight or obesity [6]. Daily avocado consumption also enhances the elasticity and firmness of the facial skin in healthy women [4]. Four key nutritional features likely mediate these beneficial effects: (1) a high unsaturated to saturated fatty acid ratio, (2) viscous, prebiotic fiber, (3) low energy density, and (4) highly bioavailable carotenoids [3]. Additionally, preclinical studies have provided preliminary evidence that various parts of the avocado have anticancer, antimicrobial, and anti-inflammatory properties [9].

It is critical to consider the intersection of complex food matrices with human physiology. A significant challenge for precision nutrition will be translating nutrient information into function. Another facet to this challenge is that we must know exactly what nutrients are in the recommended foods to deliver specific nutrition recommendations or dietary guidance. A recent comprehensive evaluation of data quality in nutrient databases highlights that “gold standard” food and nutrient databases do not provide comprehensive nutrient composition data in their current form [10]. As one stepping stone towards precision nutrition, this paper reviews the nutrient and bioactive composition of the edible Hass avocado pulp and discusses variations in nutrient composition as well as potential biological significance of select nutrients.

## 2. Unique Nutritional Physiology and Ripening of Hass Avocados

Avocados differ substantially from other fruits both physiologically and in their propagation. Avocado varieties are classified into three groups or horticultural varieties, which are named for the geographic region in which they were domesticated: (1) Guatemalan (*P. americana* var. *guatemalensis*, L.O. Williams), (2) Mexican (*P. americana* var. *drymifolia*), and (3) West Indian (*P. americana* var. *americana*) [11]. Each group has unique characteristics, including differences in leaf chemistry, peel texture and color, development period, fruit size, fruit oil content, cold hardiness, and salinity tolerance. Most of the varieties of interest for international trade are hybrids, which have been selected for fruit quality and disease and pest resistance [1]. The Hass avocado is considered a Guatemalan/Mexican hybrid because it has the thick, rough skin of the Guatemalan variety, but the high oil content of the Mexican variety [12]. The Hass avocado is commercially available in supermarkets throughout the United States and is responsible for 95% of the total commercialized volume [1].

The ripening physiology of avocados is complex. Avocado trees have a long flowering period, lasting up to 3 months. However, the percentage of flowers that become mature fruit is extremely low (<0.1%) [11]. Fruit maturation, which is the process of growth on the tree, requires 5 to 15 months after pollination [13]. Fruit can remain on the tree for more than 12 months, far beyond the time needed to reach physiological maturity, but it does not ripen on the tree [14]. Thus, at harvest, fruits of a broad range of physiological ages and maturity can be obtained from the same tree [14]. The avocado is a climacteric fruit that only starts ripening after the fruit has been picked from the tree [14].

The avocado is unique among fruits because it accumulates oil during growth and development, while most other fruits prioritize sugars. Oil accumulation begins in the mesocarp a few weeks after the flower forms a fruit, and it continues during growth and development, stopping when the fruit is harvested [11]. Oil is stored as triacylglycerols, with the primary fatty acid being oleic acid. However, the oil’s fatty acid profile varies with geographical and environmental conditions [11]. Based on dry weight, the mesocarp comprises approximately 60–70% oil and 10% carbohydrates [11].

Another unique characteristic of the avocado is that it contains large amounts of seven-carbon (C7) sugars instead of six-carbon (C6) sugars as the predominant transport and storage sugars in its leaves and fruit [11]. The sugars increase during the early stages of growth and development accompanying rapid fruit growth, and then decline as the metabolism shifts to oil accumulation [11]. C7 sugars (mannoheptulose and perseitol) inhibit the ripening process, and they may also enhance fruit quality under commercial transport and storage conditions due to their antioxidant properties [11,14].

## 3. Nutritional Composition of Ripe Hass Avocado Pulp

This review includes data from existing literature reporting nutrients in ripe avocado pulp and excludes data on unripe pulp, pulp oil, or other extracts. Data reported on a dry matter basis were converted to fresh matter weight based on the dry matter percentage noted in the original publication. If the dry matter percentage was not reported in a study, a dry matter estimate (i.e., 27.7%) calculated from the average moisture reported on United States Department of Agriculture (USDA) FoodData Central was used to project the fresh matter weight of analytes or nutrients [15]. The sample size for pooled means was determined by the number of individual analyses completed per publication or report. Pooled samples or technical replicates were considered *n* = 1. The sample size was assumed to be one if not noted in the publication.

The USDA FoodData Central is an integrated data system that provides expanded nutrient profile data that defines Hass avocados as a standard reference legacy food: “Avocados, raw, California,”; their nutrient composition data are integrated in the tables below [15]. The U.S. Nutrition Labeling and Education Act (NLEA) defines one serving of Hass avocado as 50 g (1/3 of a medium avocado), which provides 80 kilocalories, 8 g of total fat (5 g monounsaturated fatty acids), 3 g of fiber, and is a good source (≥10% dietary value (DV)) of vitamin K, folate, pantothenic acid, and copper [15].

Analytical testing of avocado pulp was also conducted with Hass avocados available for distribution in the U.S. food supply, originating from various countries and across the growing seasons. Specifically, the analysis included fresh Hass avocado sampling from May 2021, June 2021, October 2021, and December 2021. Whole fresh avocados were randomly selected and directly shipped from regional avocado packing houses to a commercial food assurance lab for ripening and analytical testing. Additionally, whole fresh avocados sampled from September 2021 were randomly selected and shipped to an independent, third-party academic lab (not affiliated with the authors of this paper) for ripening and analytical testing. Throughout this paper, these analyses are referred to as “commercial testing.” Government nutrient databases (i.e., USDA FoodData Central, USDA Special Interest Databases, Food and Drug Administration (FDA) Total Dietary Study, Australia Food Database, and New Zealand Food Composition Database) are also included in the analysis.

### 3.1. Energy and Water

The USDA reports a calculated energy content of 167 kcal/100 g avocado [15]. The pooled data from existing literature, government nutrient databases, and commercial analyses yielded a mean total energy content of 194 kcal/100 g fresh avocado pulp (median: 184 kcal/100 g; range: 138–256 kcal/100 g; n = 11) [16,17,18,19,20,21]. This corresponds to an energy density of 1.38–2.56 kcal/g. National Health and Nutrition Survey (NHANES) data shows the mean dietary energy density of the U.S. diet is 1.9 kcal/g [22]. Additionally, Hass avocados contain 61–77% water by mass. Pooled mean total water content was 72.7 g/100 g fresh avocado pulp (median = 70; range = 61–77; n = 26) [17,18,19,20,21,23,24]. Similarly, the USDA reports water content of 72.3 g/100 g avocado (range: 64–84 g/100 g; n = 33) [15].

### 3.2. Lipids

The lipid content of fresh Hass avocado pulp is shown in Table 1. The monounsaturated oleic acid (18:1n-9) is the predominant fatty acid in fresh avocados, accounting for approximately 59% of total fat based on USDA data. Palmitic acid (16:0) is the predominant saturated fatty acid accounting for 14% of total fat, while linoleic acid (18:2n-6) is the predominant polyunsaturated fatty acid, accounting for 11% of total fat (Table 1). During ripening at 20 °C, palmitic acid content decreases and polyunsaturated fatty acids increase, while monounsaturated fatty acids remain relatively unchanged [25]. However, preharvest conditions may influence fatty acid profiles. Lower growing temperatures shift the oil profile towards more oleic acid and less palmitic acid [26].

The ratio of unsaturated to saturated fatty acids in avocados is approximately 6:1. Compared with other commonly consumed fat sources in the U.S. food supply, avocados have a high proportion of unsaturated fatty acids, similar to olive oil (Figure 1). Numerous health authorities, including the 2020–2025 Dietary Guidelines for Americans, American Heart Association, and the World Health Organization, recommend consuming foods rich in monounsaturated and polyunsaturated fatty acids instead of foods high in saturated and trans-fatty acids to reduce the risk of cardiovascular disease (CVD) [27,28,29]. Observational study data suggest that replacing 5% of the energy intake from saturated fatty acids with monounsaturated fatty acids is associated with a 15% lower risk of developing coronary heart disease [30]. The recommended energy displacement could be achieved with one avocado a day. Data from the same cohorts suggest eating one or more avocados weekly was associated with a 16% and 21% reduced risk of developing CVD and coronary heart disease, respectively [31]. Research from clinical trials indicates that consuming at least one avocado per day over 4 to 5 weeks improves blood lipid profiles compared to control diets in healthy subjects with dyslipidemia and overweight or obesity [3]. Replacing saturated fatty acids with unsaturated fatty acids is also associated with a significant reduction in total mortality [32].

**Table 1 foods-12-02516-t001:** Lipids in fresh Hass avocado pulp.

	USDA Food Data Central			Literature, Other Government Databases and Commercial Analyses			
g/100 g	Mean	Min, Max	n	Pooled Mean	Min, Max	n	Refs.
Total Fat	15.4	8.4, 23.2	31	17.77	12.9, 26.7	28	[16,17,18,19,20,21,23,24]
Saturated fatty acids	2.13	NA	1	3.18	0.85, 6.3	18	[16,17,18,19,20,21,24]
16:0 Palmitic acid	2.08	1.73, 2.54	8	1.27	0.54, 4.32	121	[17,18,24,33,34]
18:0 Stearic acid	0.05	0.007, 0.082	8	0.03	0, 1.98	202	[17,18,24,34]
Monounsaturated fatty acids	9.8	NA	1	12.37	8.48, 19.51	18	[16,17,18,19,20,21,24]
16:1n-7 Palmitoleic acid	0.698	0.5, 0.881	8	0.53	0.11, 1.98	121	[17,18,24,33,34]
17:1	0.01	0, 0.016	8	0	NA	2	[18]
18:1n-9 Oleic acid	9.07	7.44, 10.9	8	4.07	1.5, 19.4	121	[17,18,24,33,34]
18:1n-7 Cis-Vaccenic acid	NA			0.627	0.35, 0.84	109	[34]
20:1n-9 Gondoic acid	0.025	0.02, 0.033	8	0.02	0.02, 0.02	2	[18]
Polyunsaturated fatty acids	1.82	NA	1	2.46	0.46, 4.55	18	[16,17,18,19,20,21,24]
18:2n-6 Linoleic acid	1.67	1.44, 1.97	8	0.93	0.29, 2.68	121	[17,18,24,33,34]
18:2n-6 Linolelaidic acid	NA			1.60	1.54, 1.66	6	[24]
18:3n-3 α-Linolenic acid	0.11	0.096, 0.128	4	0.135	0, 0.33	13	[17,18,24,33]
18:3n-6 γ-Linolenic acid	0.015	0.015, 0.015	4	0.068	0, 0.1	110	[18,34]
20:3n-6	0.016	0, 0.04	8	NA			

Based upon fresh weight sampling. NA = not applicable/available.

**Figure 1 foods-12-02516-f001:**
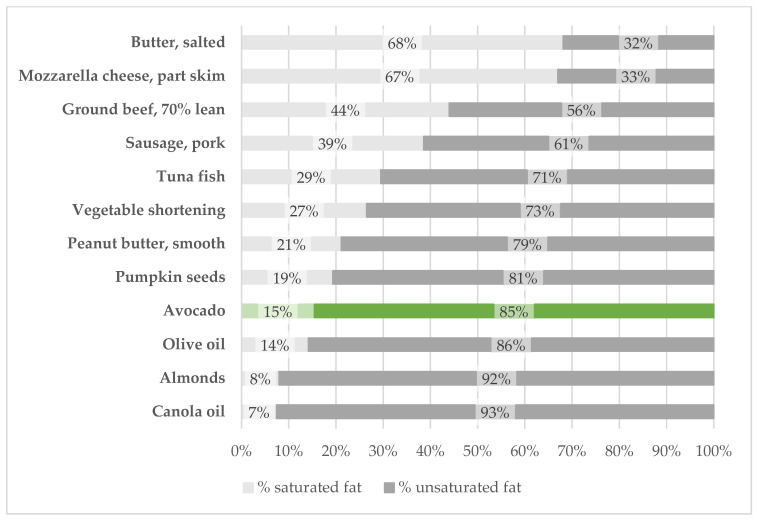
The proportion of saturated and unsaturated fatty acids of commonly consumed fat sources. Data from USDA FoodData Central [35]. FoodData Central IDs (top to bottom): 173410, 170847, 169474, 173882, 175158, 173584, 174266, 169415, 171706, 171413, 170567, 172336.

### 3.3. Carbohydrates

The carbohydrate content of fresh Hass avocado pulp is shown in Table 2. Approximately 65–80% of the carbohydrates in avocado are dietary fiber and include a mix of insoluble and soluble fibers such as cellulose, hemicellulose, and pectin [11]. The sugar content is low compared to other fruits due to their preferential oil accumulation [11].

Compared with other commonly consumed fruits and vegetables in the U.S. food supply, avocados have a high fiber content (Figure 2). The current Dietary Guidelines for Americans identify fiber as a nutrient of concern because of low consumption among the U.S. population [27]. While global guidelines generally recommend that adults consume 25–30 g of dietary fiber daily, North Americans consume an average of 17 g daily [36]. A comprehensive body of literature shows that fiber has numerous health benefits, including improvement in laxation and regularity, reduced risk of CVD moderating effects on satiety and body weight, and beneficial effects on the microbiota and gut health [36,37,38,39].

USDA FoodData Central reports 6.8 g/100 g of fiber in avocados, though this mean is over 1.5 times higher than the pooled data (3.87 g/100 g), and the amount of fiber reported is rather variable. Methodological differences in defatting or fiber analysis may contribute to this discrepancy. However, both USDA FoodData Central and the pooled data have high ranges, which suggests that sampling differences largely contribute to the high variability.

**Table 2 foods-12-02516-t002:** Carbohydrates in fresh Hass avocado pulp.

	USDA Food Data Central			Literature, Other Government Databases and Commercial Analyses			
g/100 g	Mean	Min, Max	n	Pooled Mean	Min, Max	n	Refs.
Total carbohydrates	8.64	NA	1	5.82	3, 12.2	16	[17,18,19,20,21,24]
Dietary fiber	6.8	3.2, 12.7	21	3.87	2.2, 7.5	17	[17,18,19,20,21,24]
Insoluble fiber	NA			2.63	2.56, 2.7	2	[16]
Soluble fiber	NA			2.05	1.99, 2.11	2	[16]
Total sugars	0.3	0, 0.55	11	0.1	0, 0.8	8	[18,19,20,21]
Sucrose	0.06	0, 0.15	9	0.11	0.002, 0.43	25	[33,40,41]
Glucose	0.08	0.06, 0.24	9	0.03	0.002, 0.1	22	[40,41]
Fructose	0.08	0.07, 0.15	9	0.04	0.01, 0.1	22	[40,41]
Galactose	0.08	0, 0.3	8	NA			
Starch	0.11	0.05, 0.17	4	NA			

Based on fresh weight sampling. NA = not applicable/available. Lactose and maltose were not observed.

**Figure 2 foods-12-02516-f002:**
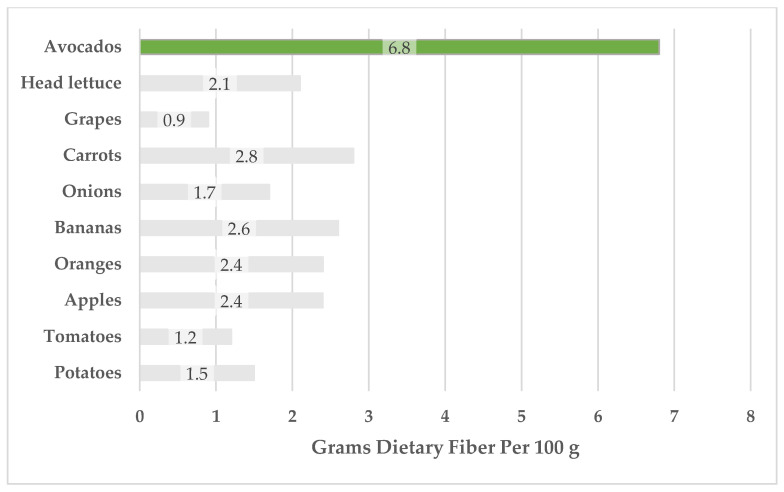
Dietary fiber in fruits and vegetables prevalent in the U.S. food supply. Food prevalence data are from USDA Economic Research Service [42], and nutrient data are from FoodData Central [15]. FoodData Central IDs (top to bottom): 171706, 169247, 174682, 170393, 170000, 173944, 169918, 171688, 170457, 170033.

### 3.4. Protein and Amino Acids

The protein content of fresh Hass avocado pulp is shown in Table 3. Although avocados contain various amino acids, they likely do not contribute a meaningful amount of protein toward daily needs.

**Table 3 foods-12-02516-t003:** Protein and amino acids in fresh Hass avocado pulp.

	USDA Food Data Central			Literature, Other Government Databases and Commercial Analyses			
g/100 g	Mean	Min, Max	n	Pooled Mean	Min, Max	n	Refs.
Total protein	1.96	1.53, 3	30	1.59	1.05, 2.4	45	[17,18,19,20,21,23,24,40]
Amino acids							
Taurine	NA			0.02	NA	1	
Hydroxyproline	NA			0.04	NA	1	
Aspartic acid	0.232		1	0.14	0.12, 0.15	2	[18]
Threonine *	0.072		1	0.07	0.06, 0.08	2	[18]
Serine	0.112		1	0.08	0.08, 0.08	2	[18]
Glutamic acid	0.28		1	0.15	0.14, 0.16	2	[18]
Proline	0.096		1	0.07	0.06, 0.08	2	[18]
Lanthionine	NA			0.04		1	
Glycine	0.102		1	0.08	0.07,0.09	2	[18]
Alanine	0.11		1	0.09	0.07, 0.1	2	[18]
Cysteine	NA			0.03	0.03, 0.04	2	[43]
Cystine	0.027		1	ND			
Valine *	0.11		1	0.09	0.08, 0.1	2	[18]
Methionine *	0.04		1	0.03	0.02, 0.04	3	[18,43]
Isoleucine *	0.08		1	0.07	0.06, 0.08	2	[18]
Leucine *	0.14		1	0.11	0.1, 0.1	2	[18]
Tyrosine	0.05		1	0.1	0.04, 0.15	2	[18]
Phenylalanine *	0.1		1	0.07	0.06, 0.07	2	[18]
Hydroxylysine	NA			0.03		1	
Lysine *	0.13		1	0.09	0.08, 0.1	2	[18]
Histidine *	0.05		1	0.03	0.03, 0.04	2	[18]
Arginine	0.09		1	0.08	0.07, 0.09	2	[18]
Tryptophan *	0.03		1	0.02	0.02, 0.02	2	[18]

Based on fresh weight sampling. NA = not applicable/available. * Essential amino acid.

### 3.5. Vitamins

One medium-sized Hass avocado provides 30% daily value (DV) vitamin K, 30% DV folate, and 45% DV pantothenic acid (Table 4). Vitamin K is vital for blood clotting, healthy bones, and other essential bodily functions. In a prospective cohort study of 56,048 participants, moderate-to-high vitamin K intake (87–192 µg/d) was linked with reduced risk of all-cause mortality, CVD-related mortality, and cancer-related mortality [44]. Per USDA FoodData Central, avocado contains 21 µg vitamin K/100 g, but values may range from 5–27 µg/100 g. Although avocado provides only one-quarter the amount of vitamin K as one cup of raw spinach (145 µg) or kale (113 µg), it also provides lipids that enhance the absorption of fat-soluble nutrients [35].

Folate is essential for maintaining a healthy pregnancy, and the 2020–2025 Dietary Guidelines for Americans identify folate as a nutrient of concern for reproductive-aged and pregnant females due to inadequate intake [27]. The recommended dietary allowance for pregnant women is 600 µg dietary folate equivalents. Pregnancy is not the only physiological condition that may require dietary folate. A meta-analysis of 12 studies involving 2570 participants found that blood folate levels were significantly lower in inflammatory bowel disease patients compared to control patients [45]. An umbrella review of 133 meta-analyses also reported the benefits of folate on all-cause mortality, multiple cancer types, CVD, and neurological conditions [46]. USDA FoodData Central reported 89 µg/100 g of avocado, consistent with the 90 µg/100 g pooled mean from the literature, commercial, and other government databases.

Pantothenic acid is a critical cofactor for energy production, specifically triacylglycerol synthesis and lipoprotein metabolism. USDA FoodData Central reports 1.46 mg/100 g pantothenic acid in avocado, almost double the amount reported in the literature, commercial, and other government databases (0.89 mg/100 g), making it one of the richest sources of pantothenic acid.

One medium-sized Hass avocado also provides 12% DV of vitamin C, 18% DV of vitamin E, 24% DV of riboflavin, and 18% DV of niacin and vitamin B6. The means from USDA FoodData Central and pooled means are relatively consistent for vitamin B6 (0.287 vs. 0.28 mg/100 g, respectively), riboflavin (0.143 vs. 0.139 mg/100 g, respectively), niacin (1.91 vs. 2.07 mg/100 g, respectively), and α-tocopherol (1.97 vs. 2.13 mg/100 g, respectively). The means for the pooled vitamin C data are 6.19 mg/100 g with a range between 1.9–13 mg/100 g, suggesting sample or method variability may contribute to the differences between these data and USDA FoodData Central (8.8 mg/100 g).

**Table 4 foods-12-02516-t004:** Vitamins in fresh Hass avocado pulp.

	USDA Food Data Central			Literature, Other Government Databases and Commercial Analyses			
Per 100 g	Mean	Min, Max	n	Pooled Mean	Min, Max	n	Refs.
Vitamin C (mg)	8.8	6.3, 13.9	16	6.19	1.9, 13	15	[16,17,18,19,20,21,41]
Thiamin (mg)	0.075	0.052, 0.1	12	0.069	0.03, 0.119	10	[17,18,19,20,21]
Riboflavin (mg)	0.143	0.119, 0.18	12	0.139	0.12, 0.183	10	[17,18,19,20,21]
Niacin (mg)	1.91	1.46, 2.51	12	2.07	1.59, 2.6	10	[17,18,19,20,21]
Pantothenic acid (mg)	1.46	0.93, 2.71	12	0.89	0.65, 1.2	6	[17,18,19,20,21]
Pyroxidine (mg)	0.287	0.196, 0.452	11	0.28	0.1, 0.69	10	[17,18,19,20,21]
Folate (µg)	89	71, 155	20	90	61,120	10	[17,18,19,20,21]
Biotin (µg)	NA			2.73	0, 5.6	6	[17,18,19,20,21]
Vitamin A * (µg)	7	NA	NA	10.5	6, 16	4	[18,19,20,21]
α-tocopherol (mg)	1.97	0.66, 3.28	22	2.13	0.94, 3.28	20	[17,18,41,47]
β-tocopherol (mg)	0.04	0.02, 0.06	9	0.01	0, 0.05	5	[18]
γ-tocopherol (mg)	0.32	0.09, 0.75	18	0.25	0, 0.75	14	[18,47]
δ-tocohpherol (mg)	0.02	0.01, 0.03	9	0.03	0,0.13	9	[18,41]
Vitamin K (µg)	21	15.7, 27	8	16.55	5, 25	6	[19,20,21]
Other							
Choline (mg)	14.2	NA	NA	19.5	19.3, 19.6	2	[17]

Based on fresh weight sampling. NA = not applicable/available. * Retinol activity equivalent.

### 3.6. Minerals

The mineral content of fresh Hass avocado pulp is shown in Table 5. One medium-sized Hass avocado (150 g) provides 30% DV copper and 18% DV potassium. One avocado provides nearly 100% of the daily needs of copper for infants (0.2–0.22 mg). Copper is a cofactor for energy production pathways, iron metabolism, synthesis of connective tissues, lipid metabolism, and activation of neuropeptides [48]. According to national survey data, about 30% of the general U.S. population does not meet recommended copper intake levels, which has been hypothesized to contribute, in part, to dyslipidemia [48].

Potassium is critical in managing blood pressure [49]. Dietary potassium intake of at least 2900 mg/d is associated with a reduced incidence of type 2 diabetes [50], and intake >4000 mg/d is associated with a reduced risk of developing kidney stones [51,52]. The 2020–2025 Dietary Guidelines for Americans identifies potassium as a nutrient of concern because of low intakes across the U.S. population [27].

**Table 5 foods-12-02516-t005:** Minerals and trace minerals in fresh Hass avocado pulp.

	USDA Food Data Central			Literature, Other Government Databases and Commercial Analyses			
Per 100 g	Mean	Min, Max	n	Pooled Mean	Min, Max	n	Refs.
Calcium (mg)	13	8, 19	24	11.7	8, 15	43	[17,18,19,20,21,24,53]
Iron (mg)	0.61	0.29, 1.06	34	0.65	0.4, 2.3	43	[17,18,19,20,21,24,53]
Magnesium (mg)	29	19, 34	12	30.64	19, 64	43	[17,18,19,20,21,24,53]
Phosphorus (mg)	54	41, 70	12	44.0	26.3, 55	43	[17,18,19,20,21,24,53]
Potassium (mg)	507	356, 691	24	478.0	408, 1010	44	[16,17,18,19,20,21,24,53]
Sodium (mg)	8	2, 17	18	3.57	1.5, 18	43	[17,18,19,20,21,24,53]
Zinc (mg)	0.68	0.49, 0.83	12	0.52	0.35, 1.1	41	[18,19,20,21,24,53]
Copper (mg)	0.17	0.09, 0.38	12	0.25	0.15, 0.34	43	[17,18,19,20,21,24,53]
Manganese (mg)	0.149	0.106, 0.19	12	0.17	0.08, 0.4	43	[17,18,19,20,21,24,53]
Selenium (µg)	0.4	0.2, 0.6	5	0.1	0, 0.9	31	[18,19,20,21,53]
Fluoride (µg)	NA			230		1	[18]
Iodine (µg)	NA			0.08	0, 1.5	31	[18,19,20,21,53]
Nickle (mg)	NA			0.03	0, 0.21	30	[17,18,53]
Chloride (mg)	NA			30		1	[18]
Chromium (mg)	NA			0.001	0, 0.018	30	[17,18,53]
Molybdenum (µg)	NA			0.0003	0.0002, 0.0003	28	[18,53]
Silicon (mg)	NA			31	10, 51	2	[17]
Boron (mg)	NA			3.7	2.6, 4.8	2	[17]
Strontium (mg)	NA			0.15	0.11,0.97	29	[17,53]

Based on fresh weight sampling. NA = not applicable/available.

## 4. Bioactive Compounds in Hass Avocado Pulp

A literature review of avocado compounds, data collected from USDA FoodData Central, USDA Special Interest Database, the Australian Food Standards, and commercial analysis was compiled. Pooled means from all sources were calculated for each compound, and minimum and maximum values are indicated from available sources (Table 6, Table 7, Table 8, Table 9, Table 10 and Table 11). Not all sources reported minimum or maximum values. Primary classes include fatty alcohols, phenolic compounds, organic acids, carotenoids, sterols, etc. Literature also reported the detection of some compounds without quantification (e.g., qualitative mass spectroscopy).

### 4.1. Fatty Alcohols

Avocados contain several unique long-chain fatty alcohols with bioactive properties. Persenone A, personone B, and acetylated-avocadene are the fatty alcohols with the highest concentrations (Table 6). These compounds are also classified as acetogenins (Figure 3), which are associated with anticancer and proapoptotic activity.

Early research on avocado persenones suggested they may be bioactive. In vitro studies indicate that persenone A and B may inhibit superoxide and nitric oxide, thus, suppressing radical generation [54]. In another study, authors estimated that a dose of 25 mg persenone A per kg body weight would attenuate the formation of thrombi [55]. Persenone C was shown to possess antiplatelet activity in vitro, while in vivo, persenone A demonstrated potential protective effects against arterial thrombosis with increased coagulation time [56]. The researchers hypothesized that acetogenins might be responsible for previous research results reporting that avocado intake significantly lowered platelet aggregation levels (∼30%) compared with patients who did not consume avocado [57]. Lastly, in vitro avocado extracts containing persenone A and C were shown to inhibit *Listeria monocytogenes*. The total content in pulp was 199 to 398 times higher than the required minimum inhibitory values. Therefore, avocado consumption can provide adequate acetogenin levels to inhibit *Listeria* [58].

Avocado pulp also contains the polyhydroxylated alcohols avocadyne and avocadene. In vitro, avocadyne was shown to be a potential inhibitor of fatty acid oxidation. Together, these molecules may antagonize each other’s actions. Additionally, avocadyne was shown to spare normal hematopoietic cells while suppressing primary acute myeloid leukemia cell growth and reducing cell engraftment in vivo [59]. Further mechanistic studies indicate that avocadyne targets very long acyl-CoA dehydrogenase switching toward glycolysis leading to leukemic cell death [60]. Avocatin B, a 1:1 mixture of avocadyne and avocadene, induced the death of leukemia cell lines and patient-derived acute myeloid leukemia cells [61]. Moreover, supplementation with avocatin B improved glucose tolerance, glucose utilization, and insulin sensitivity in preclinical mouse models [62].

**Figure 3 foods-12-02516-f003:**
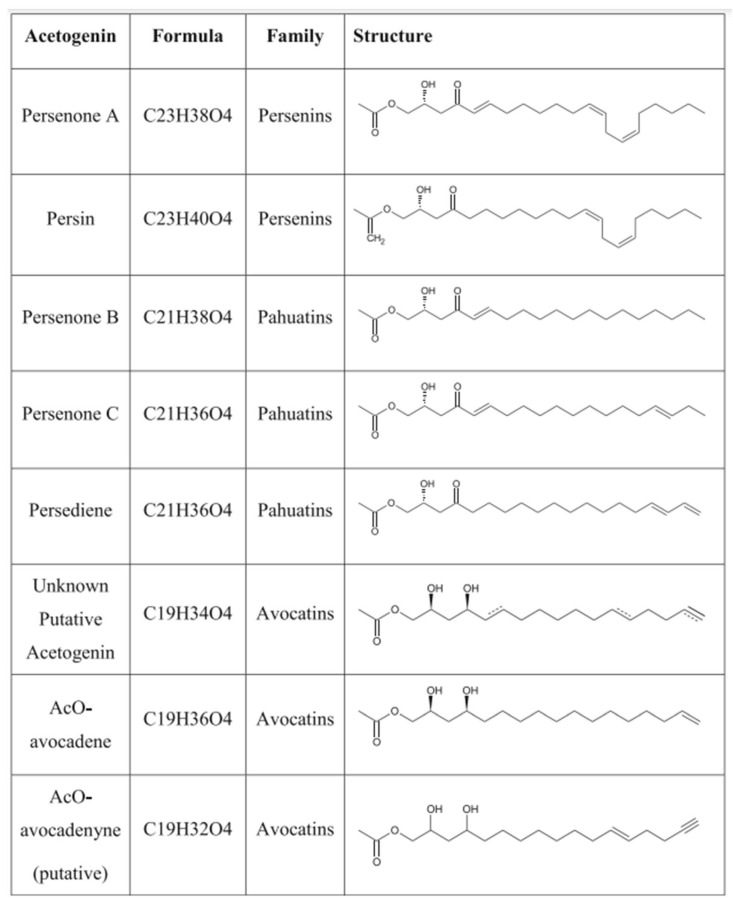
Structure of acetogenins. Reproduced with permission from: Rodríguez-López et al. [13]. Avocado fruit maturation and ripening: dynamics of aliphatic acetogenins and lipidomic profiles from mesocarp, idioblasts, and seed. *BMC Plant Biol* 17, 159 (2017).

**Table 6 foods-12-02516-t006:** Fatty alcohols in fresh Hass avocado pulp.

mg/100 g	Pooled Mean	Min, Max	n	Refs.
Avocadyne	4.99 *	NA	3	[63]
Avocadene	6.09 *	NA	3	[63]
Personone A	172.5	163, 182	6	[13,64]
Persenone B	56.5	34, 79	6	[13,64]
Persenone C	31	26, 36	6	[13,64]
Persediene	4	NA	3	[13]
Acetylated-avocadene	1	NA	3	[13]
Acetylated-avocadyne	40.5	30, 51	6	[13,64]
Avoenin	0.98 *	NA	1	[65]

Based on fresh weight sampling. NA = not applicable/available. * Includes projected fresh weight data based on 27.7% dry weight.

### 4.2. Seven-Carbon Carbohydrates

Carbohydrates within the avocado pulp are primarily fibers; however, unique C7 carbohydrates have also been identified and quantified (Table 7). Mannoheptulose (9–347.6 mg/100 g) and its polyol form, perseitol (16–424.2 mg/100 g), comprise the majority of C7 carbohydrates in the avocado pulp. These reports have high variability, likely due to their rapid ripening utilization. Storage at 20 °C for three days resulted in a 50% lower content of mannoheptulose [25]. Blakey et al. [40] reported that the rate of ripening impacted mannoheptulose concentrations within the pulp, with slower ripening resulting in a greater amount of mannoheptulose. Seasonality also affects C7 levels. For example, unripe, early-season fruit in Spain had nearly 5-fold greater levels of mannoheptulose (356 g/100 g) than unripe late-season fruit [25].

Mannoheptulose is a well-known inhibitor of hexokinase and glucokinase—enzymes responsible for phosphorylating glucose in the first step of glycolysis [66]. Inhibition of these enzymes inhibits glycolysis and suppresses insulin secretion [66]. High doses of intravenous mannoheptulose markedly increase blood glucose levels while reducing serum insulin [67]. However, lower oral doses (e.g., from dietary avocados) reduced serum insulin concentrations without altering blood glucose concentrations [68,69]. In a study of eight healthy volunteers, participants consumed as much partially ripe avocado as they could comfortably eat after an overnight fast. The amount of avocado eaten ranged from 184 to 570 g, corresponding to a mannoheptulose dose of 2.15 to 12.83 g per person or 33 to 200 mg/kg body weight [69]. Although no changes in blood glucose were observed, circulating insulin concentrations were reduced [69].

Preclinical studies suggest the role of mannoheptulose as a glycolytic inhibitor may also function as an energy restriction mimetic. Hexokinase inhibition stimulates cellular responses that mimic energy restriction [68]. In vitro, a mannoheptulose-enriched avocado extract increased the expression of genes associated with beneficial effects of energy restriction, including phosphorylated AMPK, AMPKα1, AMPKα2, Sirt1, and PGC-1α [68]. Furthermore, increased fatty acid oxidation was also observed [68]. In mice, mannoheptulose given at 1.7 g/kg body weight resulted in protective effects against diet-induced obesity [70]. However, this dose is much higher than what can be achieved from eating fresh avocados.

**Table 7 foods-12-02516-t007:** C7 carbohydrates in fresh Hass avocado pulp.

mg/100 g	Pooled Mean	Min, Max	n	Refs.
Mannoheptulose	240.2	9, 347.6	39	[25,33,40,41]
Perseitol	292.9	16, 424.2	39	[25,33,40,41]
Volemitol	1.06	NA	4	[41]

Based on fresh weight sampling. NA = not applicable/available.

### 4.3. Phenolics and Organic Acids

The total phenolic content in fresh avocado pulp ranges from 1–26 mg/100 g (Table 8). Although low relative to other food sources, epigallocatechin is the predominant phenolic compound identified. The organic acid reported in the highest content was succinic acid (1173 mg/100 g). Succinate, the conjugate base of succinic acid, is a metabolic intermediate in the TCA cycle, and a metabolic signaling molecule in inflammation processes [71].

Numerous other phenols and organic acids are present as well. Phenolic acids do not participate in the primary metabolism in the avocado, but they are involved in the secondary metabolism as precursors to other phenolic compounds and cell-wall components [11]. The organic acid content of fresh avocado pulp is shown in Table 9. Organic acids are essential intermediates in plant metabolic pathways. They can act as precursors for synthesizing amino acids, plant hormones, fatty acids, secondary metabolites, and cell-wall components [11]. Little is known about the role of organic acids during growth, development, and ripening; however, total acids tend to decrease as the fruit ripens [11]. Organic acids, including ascorbic acid, can form complexes with minerals, aiding in bioavailability. Organic acids are less potent compared to ascorbic acid in aiding digestibility. For example, evidence suggests 1000 mg of citric acid to 3 mg of iron would enhance nonheme iron absorption [72]. Avocados contain fewer organic acids than other fruits, for which organic acids are essential contributors to fruit quality [11]. With this ratio in mind, a medium-sized avocado (150 g each) contains about 324 mg of citric acid, which may enhance the absorption of about 1 mg of iron (equates to about 5% DV).

**Table 8 foods-12-02516-t008:** Phenolic content of fresh Hass avocado pulp.

mg/100 g	Pooled Mean	Min, Max	n	Refs.
Total Phenolic Content (GAE)	20 ^§^	13.3, 26	15	[23,41,73]
Total Phenolic Content	6.1 *	1.5, 10.7	2	[74,75]
Epicatechin	0.48	0.08, 1.11	12	[74,76,77]
Epigallocatechin	1.03	0.96, 1.1	2	[76]
Cyanadin	0.5	0.42, 0.58	4	[76]
Nargenin	0.007	NA	1	[77]
Quercetin	0.557	NA	1	[77]
Rutin	0.006	NA	1	[74]
Taxifolin	0.005	NA	1	[74]
Vanillin	0.002	NA	1	[74]
4-hydroxybenzoic acid	0.02	0.005, 0.03	2	[74,77]
Caffeic acid	0.02	NA	1	[74]
Caffeic acid glucoside	0.27 *^,§^	NA	1	[75]
Chlorogenic acid	0.015	0.006, 0.023	2	[74,77]
Ferulic acid	0.19	0.15, 0.23	2	[74,77]
Ferulic acid glucoside isomers	0.75 *^,§^	NA	1	[75]
3-feruloylquinic acid	0.21 ^§^	NA	1	[75]
5-feruloylquinic acid	2.11 ^§^	NA	1	[75]
4-feruloylquinic acid	0.22 ^§^	NA	1	[75]
Gentisic acid	0.02	NA	1	[74]
Isoramnetin	0.003	NA	1	[74]
Coumaric acid	0.64 *	0.47, 0.82	2	[75,77]
*p*-coumaric acid	0.58	0.36, 0.79	2	[74,77]
*m*-coumaric acid	0.032	NA	1	[77]
*p*-coumaric acid glucoside isomers	2.62 *^,§^	NA	1	[75]
*p*-coumaric acid pentoside	0.29 *^,§^	NA	1	[75]
*p*-coumaric acid rutinoside	0.45 *^,§^	NA	1	[75]
Sinapic acid-C-hexoside	0.21 *^,§^	NA	1	[75]
Sinapic acid	0.03	NA	1	[77]
Tyrosol-hexoside-pentoside	0.63 *^,§^	NA	1	[75]
Octyl gallate	0.26 *^,§^	NA	1	[75]
Trans-cinnamic acid	0.052	0.005, 0.98	2	[74,77]
Sinapinic acid	0.04	NA	1	[73,74]

Based on fresh weight sampling. GAE (gallic acid equivalent). NA = not applicable/available. * Includes projected fresh weight data based on 27.7% dry weight. ^§^ Value represents an estimation of actual concentrations based on a standard curve of a structurally similar compound. The following compounds have been detected in the avocado pulp with mass spectroscopy: 4′-O-Methyl-(-)-epigallocatechin 7-O-glucuronide, 2-Hydroxy-2-phenylacetic acid, 3-Hydroxyphloretin 20-O-glucoside, Phloridzin, Myricetin 3-O-arabinoside, Myricetin 3-O-rhamnoside, 20-Hydroxyformononetin, 7-Oxomatairesinol, 40-Hydroxy-3,4,5-trimethoxystilbene, 2-Hydroxy-4-methoxyacetophenone 5-sulfate, Coumarin, 2-Hydroxybenzoic acid, Protocatechuic acid 4-O-glucoside, Galloyl glucosa, Isoferulic acid, Cinnamic acid, 3-Sinapoylquinic acid, 1-Sinapoyl-2-feruloylgentiobiose [73].

Polyphenols are a large class of over 8000 naturally occurring compounds, many of which have antioxidant properties. They are secondary plant metabolites that protect against ultraviolet radiation and pathogens [78]. In foods, they may contribute to the color, flavor, odor, bitterness, astringency, and oxidative stability [78]. There are four main classes of polyphenols: phenolic acids, flavonoids, stilbenes, and lignans. Epicatechin is a flavonol that is found at high levels in tea, cocoa, grapes, and apples [79]. Data from pre-clinical studies indicate that epicatechin has antioxidant and anti-inflammatory properties. It may enhance skeletal muscle performance, modulate insulin signaling pathways, and have cardioprotective, neuroprotective, and anticancer effects [79].

**Table 9 foods-12-02516-t009:** Organic acids in fresh Hass avocado pulp.

mg/100 g	Pooled Mean	Min, Max	n	Refs.
Succinic acid	1 *^,§^	0.3, 1.2	5	[41,75]
Fumaric acid	26.9	NA	4	[41]
Quinic acid	20.6 *^,§^	0.03, 30.9	6	[41,75,77]
Malic acid	119.2	NA	4	[41]
Citric acid	216 *	NA	5	[41,75]
Oxalic acid	ND		4	[41]
Benzoic acid	0.11	0.1, 0.13	2	[74,77]
Abscisic acid	0.267	NA	1	[77]
Homovanillic acid	0.002	NA	1	[77]

Based on fresh weight sampling. NA = not applicable/available. * Includes projected fresh weight data based on 27.7% dry weight. ^§^ Value represents an estimation of actual concentrations based on a standard curve of a similar compound.

### 4.4. Carotenoids and Other Pigments

Carotenoids are a diverse group of secondary metabolites critical for plant growth and development produced only in plants, algae, fungi, and bacteria. Colorless carotenoids are metabolized to the red pigment, lycopene, which is metabolized to carotenes, xanthophylls, and other compounds [80]. Carotenes and β-cryptoxanthin are metabolized to retinol. The most common carotenoids consumed in U.S. diets are alpha-carotene, β-carotene, β-cryptoxanthin, lutein, zeaxanthin, and lycopene [81]. The pooled carotenoid means were calculated from three papers, USDA FoodData Central, the Australian Food Standards, and commercial testing (Table 10). Neoxanthin (448 µg/100 g), lutein (514 µg/100 g) and its metabolite lutein-5,6-epoxide (402 µg/100 g), and violaxanthin (202 µg/100 g) are the carotenoids with the highest amount in the avocado pulp. However, the range for these carotenoids is relatively large. For example, the variability of lutein is 140–842 µg/100 g.

Many factors can influence carotenoid content and contribute to variability in concentration and absorption. The sampling location within the flesh of the Hass avocado considerably impacts carotenoid measurements, with total carotenoids found at the highest amount in the dark green flesh closest to the skin [82]. All carotenoids decrease postharvest when maintained at 20 °C, with the most substantial declines in neoxanthin [82]. Hass avocado carotenoid content also strongly correlates with total fat content [83], which can vary (as detailed in Section 3.2). The fruit ripening stage affects the bioavailability of fat-soluble carotenoids from avocado fruit [84]. Total fat, type of fat, presence of soluble fiber, food processing, and phytosterols can affect carotenoid absorption [85,86]. Avocado consumption has been shown to enhance carotenoid absorption primarily due to its lipid content [87,88].

Neoxanthin and violaxanthin are primarily known for their photoprotective roles in plants [89,90]. However, relatively little is known about their potential health effects in humans. Early evidence from animal models and in vitro studies suggests that neoxanthin may have anticancer [91,92,93], anti-adipocyte differential properties [94], and anti-inflammatory properties [95]. In vitro studies also suggest violaxanthin may have anticancer properties [96,97].

Lutein has been studied for its role in eye and skin health and cognition. Although carotenoids accumulate in tissues throughout the body, lutein and zeaxanthin preferentially accumulate in the eye and brain, while other carotenoids are deposited primarily in adipose and liver tissue [81,98]. Lutein is mainly concentrated in the central area of the retina, known as the macula. It is believed to protect against harmful blue light, oxidative damage, and macular degeneration. Increased concentration of lutein in the macula, as measured by macular pigment optical density (MPOD), contributes to enhanced visual performance, reduced visual discomfort, reduced glare sensitivity, and improved contrast sensitivity [98]. Randomized controlled trials have shown the beneficial effects of lutein supplementation on visual function at doses ranging from 8 mg to 40 mg per day [99,100]. To achieve this level of lutein intake from avocado, one must consume 11–15 medium-sized Hass avocados (150 g each). Data from observational studies have linked higher intakes of lutein with a reduced risk of developing age-related macular degeneration; however, the results have been inconsistent [101]. Lutein’s beneficial effects on visual performance are believed to be mediated by its biochemical antioxidant and physical blue-light filtering properties [98].

Lutein also preferentially accumulates in neural tissue and is the major carotenoid in the brain despite not being the predominant carotenoid in the serum [99]. MPOD may serve as a biomarker for brain lutein as lutein in the macula significantly correlates with levels in matched brain tissue [99]. Higher MPOD is associated with improved cognitive function in older adults [98]. A randomized placebo-controlled clinical trial found that 12 mg lutein + zeaxanthin daily for one year significantly increased MPOD and cognitive improvements in older adults [102]. Daily lutein (12 mg) with or without docosahexaenoic acid for four months improved cognitive function in older women compared to placebo [103]. A randomized clinical trial found that a 6-month dietary intervention with one avocado daily increased serum lutein, MPOD, and improved cognitive function in older adults [7]. However, the same effects were not seen in young to middle-aged adults with overweight or obesity [5].

As lutein and other carotenoids accumulate in the epidermis and subcutaneous adipose tissue, its antioxidant and blue-light filtering properties may benefit skin health. A randomized, placebo-controlled trial found that oral supplementation of lutein and zeaxanthin (12 mg/day for 12 weeks) improved skin tone and increased the UV threshold dose that produced sunburn [104]. Another randomized, placebo-controlled trial found that an oral supplement containing both lutein (5 mg) and zeaxanthin (0.3 mg) increased measures of photoprotection, skin elasticity, and skin hydration while decreasing measures of lipid peroxidation when taken twice daily for two weeks [105]. While these doses are higher than what can be obtained from eating an avocado, a randomized clinical trial found that daily avocado consumption enhances the elasticity and firmness of the facial skin in healthy women [4].

**Table 10 foods-12-02516-t010:** Carotenoids and other pigments in fresh Hass avocado pulp.

	USDA Food Data Central			Literature, Other Government Databases and Commercial Analyses			
µg/100 g	Mean	Min, Max	n	Pooled Mean	Min, Max	n	Refs.
Lutein and Zeaxanthin	271	170, 379	16	541	223, 874	209	[41,47,83]
Lutein	NA			514	140, 842	224	[41,47,82,83]
Zeaxanthin	NA			8	1, 100	209	[41,47,83]
β-cryptoxanthin	27	0, 120	25	23	17, 64	206	[18,47,83]
Neoxanthin	NA			448	46, 1190	192	[83]
Lutein-5,6-epoxide	NA			402	2, 899	196	[41,83]
9′-cis-neoxanthin	NA			102	6, 216	196	[41,83]
cis-violaxanthin	NA			202	44, 475	192	[83]
Neochrome	NA			96	37, 161	192	[83]
Chrysanthemaxanthin	NA			159	31, 272	192	[83]
15-cis-zeaxanthin	NA			13	NA	4	[41]
13-cis-lutein	NA			6	NA	4	[41]
15-cis-lutein	NA			36	NA	4	[41]
Alpha-carotene	24	0, 100	27	40	3, 89	206	[18,47,83]
Chlorophyll a	NA			1.84	NA	1	[41]
Chlorophyll b	NA			1.16	NA	1	[41]
Pheophorbide a	NA			0.006	NA	1	[41]
Pheophytin a	NA			0.015	NA	1	[41]

Based on fresh weight sampling. NA = not applicable/available.

### 4.5. Phytosterols

Phytosterols are compounds structurally related to cholesterol and are classified as plant sterols or stanols. β-sitosterol, campesterol, and stigmasterol are the most common plant-derived sterols in the human diet [106]. The pooled means are presented in Table 11. Six sterols were detected and reported as milligrams of phytosterol per 100 g of avocado pulp. There are three predominant phytosterols: β-sitosterol, cycloartenol, and campesterol. β-sitosterol is the most abundant phytosterol in avocado—USDA FoodData Central and the pooled means reported means of 76 and 57 mg/100 g, respectively. The variance in the reported values is high for β-sitosterol from 24–105 mg/100 g.

Phytosterols are well-known to lower LDL-cholesterol levels by inhibiting intestinal cholesterol absorption and decreasing hepatic synthesis [106,107]. The FDA currently allows the use of the following authorized health claim: “Foods containing at least 0.65 g per serving of plant sterol esters, eaten twice a day with meals for a daily total intake of at least 1.3 g, as part of a diet low in saturated fat and cholesterol, may reduce the risk of heart disease” [108]. Fifteen avocados would be required to achieve this level of daily intake. While a phytosterol intake of up to 600 mg/day can be reached from natural sources with vegan or vegetarian diets, this is still less than half the required amount of phytosterols to reduce coronary heart disease [108,109]. As the FDA recommends, adequate intake can only be achieved through foods specifically enriched in phytosterols, such as margarines, juices, or supplements [109].

Some evidence suggests that phytosterols play a role in prostate health. Benign prostatic hyperplasia (BPH), an enlargement of the prostate gland, is a common condition in older men that can interfere with urination [110]. Berges et al. [111] conducted a randomized, double-blind, placebo-controlled trial evaluating 60 mg β-sitosterol per day in 200 patients with BPH. After six months of treatment, improvements were seen using the International Prostate Symptom Score and Boyarsky quality of life score [111]. β-sitosterol treatment also increased peak urinary flow and decreased mean residual urinary volume compared with placebo [111]. An 18-month follow-up study found that these improvements were maintained in patients who continued β-sitosterol treatment [112].

Similarly, Klippel et al. [113] conducted a randomized, double-blind, placebo-controlled trial evaluating 130 mg β-sitosterol daily in 177 patients with BPH for six months. β-sitosterol treatment improved International Prostate Symptom Score scores and the Boyarsky quality of life score, increased the peak urinary flow, and decreased the residual urine volume compared with the placebo [114]. A systematic review that included additional studies found that β-sitosterol treatments were well tolerated and improved urinary symptoms and flow in men with mild to moderate BPH [110]. To achieve the level of sterol intake used in these studies from avocados only, roughly ½-1 medium avocado (150 g each) would need to be consumed daily.

**Table 11 foods-12-02516-t011:** Phytosterols in fresh Hass avocado pulp.

	USDA Food Data Central			Literature, Other Government Databases and Commercial Analyses			
mg/100 g	Mean	Min, Max	n	Pooled Mean	Min, Max	n	Refs.
β-sitosterol	76	62, 98	6	57	24, 105	85	[34,41,114]
Stigmasterol	2	2, 2	6	0.94	0.14, 10	85	[34,41,114]
Campesterol	5	5, 6	6	6	4, 11	85	[34,41,114]
Cycloartenol	NA			17	NA	4	[41]
Avenasterol	NA			3.9	NA	8	[114]
Stanol	NA			0.5	NA	8	[114]

Based on fresh weight sampling. NA = not applicable/available.

### 4.6. Glutathione and Betaine

Fresh Hass avocado pulp also contains betaine and glutathione. USDA FoodData Central [35] reported 0.7 g/100 g of betaine; however, it did not report on measures of glutathione. Jones et al. [43] reported 27.7 g/100 g glutathione in avocados, one of the highest concentrations of glutathione among fruits and vegetables [115]. Glutathione is a tripeptide cofactor for the antioxidant enzyme glutathione peroxidase. Although plasma glutathione is broadly linked to health outcomes such as cardiometabolic health [116], mental health [117,118], and reduced oxidative stress, the impact of dietary glutathione on health outcomes is inconclusive, likely due to the variability in absorption, digestion, and food preparation [115]. However, a 6-month, randomized controlled trial found that 250 mg/day of oral glutathione decreases markers for oxidative stress compared to the placebo group [119]. To achieve the dose in this study, about six medium avocados (150 g each) must be consumed daily.

## 5. Challenges in Moving toward Precision Nutrition

Multiple barriers complicate precision nutrition, including challenges with compound data extraction from publications, analytic method variation, and translating food nutrient data to function in humans. Although not a unique challenge with avocados, these barriers as well as the natural variability in agricultural practices adds to the challenge of precision with fresh products [120].

Data extraction challenges include finding and accessing information on ripe fruit and reporting variability. Finding the data to be included in this paper was challenging because reports were archived in multiple databases or were behind paywalls. PubAg (https://pubag.nal.usda.gov/ (accessed on 1 February 2023)) identified the greatest number of relevant publications for developing this dataset. Another constraint was that data from many nutrients and compounds presented in this paper were derived from few sources, which limits confidence in the data. Additional work on these and other avocado-derived nutrients is warranted. The USDA FoodData Central SR Legacy foods database, which includes Hass avocados, currently identifies up to 149 nutrients and compounds. In contrast, plants may produce more than 200,000 metabolites and many additional secondary metabolites [10], suggesting a large number of compounds have yet to be identified and/or quantified in the avocado. Additionally, more information is needed on the dietary fiber in avocados. Data on soluble and insoluble fiber is sparse, and to our knowledge, there have been no publications to date measuring the types of fibers present in avocado.

Another challenge with collating the acquired publications and data was interpretation due to a lack of standardized data reporting. For instance, data were reported as dry or wet weight, further complicating comparisons between manuscript data. Data reported as % oil, residue, or extracts were excluded because methodological differences limited the translatability to the nutrient value of the whole fresh fruit. If the information to calculate 100 g fresh weight was not available, those publications were excluded by the metadata pooled means.

The USDA FoodData Central data analyses for Hass avocado span 35 years from 1982 to 2017. Sample sizes are also relatively small for many nutrients across the reporting period. Analytical methods have dramatically changed over time and may be responsible for some of the variability reported in data as different methods give different yields leading to variable reports.

Agricultural growing practices and postharvest handling of fruit strongly impact nutrient concentration. Seasonal variation, how long the fruit stays on the tree, and growing conditions affect fruit maturity [17,83]. As the fruit matures, total fat, oleic acid, fiber, energy density, potassium, and vitamin E increase while water, carbohydrates, saturated fatty acids, polyunsaturated fatty acids, and vitamin C decrease. Changes in postharvest ripening of fatty acid content are too small to be significant in precision nutrition [121]. On the other hand, the consumer has little to no control of most pre- or post-harvest factors that may influence nutrient composition. However, one practical consideration when preparing fresh fruit is to scoop or scrape near the peel to retain the darker green flesh near the peel, where carotenoids are the highest [82].

Moreover, the dry matter requirements to bring avocado fruits to market have changed over time (21–28%), and the mode of transport (truck vs. boat; refrigeration and ethylene gas exposure) and length of the trip determine when the fruit is picked to ensure avocados reach the marketplace with a desirable dry matter and oil content. Fruit hydration/dehydration throughout the season, during physical transport, and with ripening drastically affects the percent oil content. Avocados are unusual in that 98% of the avocado fruit cells have very similar nutritional properties. Thus, fruit size contributes minimally to nutrient variability [122]. Precision agricultural practices focus on maximizing fruit production and minimizing agricultural inputs, but future research could also prioritize maximizing nutrient content and availability in the Hass avocado.

A significant challenge for precision nutrition will be translating avocado nutrient information to function. Beyond the nutrient composition of the fresh avocado, it is also important to characterize the digestion of the avocado food matrix, absorption of nutrients and compounds, and transport to target tissues in humans. For example, we might speculate that vitamin C and organic acids in avocados would enhance iron and calcium absorption but may decrease the absorption of other minerals like copper. Additionally, data from other foods show that processing and how bioactive molecules are found within the food matrix impact the metabolizable energy from foods [123]. Thus, we may extrapolate this to avocados. However, no data exist yet to substantiate these hypotheses, and additional research is necessary to define the nutrient bioavailability and their impact on the biological targets described throughout this paper.

Moreover, precision nutrition involves more than an individual food. For instance, health effects occur in response to consuming whole fresh avocado, often as part of a meal, not as an individual food or nutrient. When eaten alongside other foods, avocados can act as a nutrient booster and help absorb fat-soluble nutrients found in those foods. Two postprandial studies demonstrated that when an avocado is consumed with other foods, carotenoid absorption is maximized, and conversion to vitamin A is enhanced, especially in populations with retinol deficiency [88]. Avocados alone, however, may not be able to compensate for other dietary and lifestyle choices. In the Habitual diet and Avocado Trial (HAT), where 1008 participants were randomized to consume a large avocado daily or maintain their habitual diet for six months, researchers found modest reductions in total cholesterol and LDL-cholesterol [124]. In contrast, much more substantial reductions in LDL-cholesterol and non-HDL-cholesterol were observed in a controlled feeding study that replaced food sources of saturated fatty acids with avocado [125]. Although the diet quality increased with both studies, there was no change in cholesterol or saturated fatty acid intake in the HAT study, suggesting that simply adding avocado to the diet without making swaps provides less benefit on these cardiometabolic risk factors.

Lastly, the unique physiological needs of different populations across the lifespan and with varying metabolic conditions and health would further challenge the interpretation of how fresh Hass avocados may impact health. Comparing the Wang et al. [125] and Lichtenstein et al. [124] studies again, the avocado intervention impacted more dramatically cholesterol profiles in the study with participants with elevated LDL-cholesterol at baseline [124,125]. Furthermore, a recent meta-analysis confirms avocado intake may only reduce total cholesterol and LDL-cholesterol in people with hypercholesterolemia [126]. There may also be benefits from avocado consumption patients suffering from metabolic dysfunction, in particular those with elevated mitochondrial fatty acid oxidation (FAO). Avocatin B targeted mitochondria and inhibited FAO in two disease states: leukemia and insulin resistance. In both pathologies, avocatin B inhibited FAO. However, in leukemia, this metabolic targeting resulted in selective cancer cell death with no effect in normal cells [61]. In contrast, in skeletal muscle or insulin-producing β-islet cells, this shifted metabolic substrate utilization toward glucose uptake and utilization and insulin sensitivity [62]. Given nutrient variability, the complexity of the food matrix, and nutrient interactions within a meal, more research is needed to understand the effect of consuming whole fresh avocados on health outcomes in individuals with cancer or diabetes.

## 6. Conclusions

Precision nutrition faces many challenges. Avocados are unique compared to other foods due to their high unsaturated to saturated fat content, low carbohydrate levels, high dietary fiber, and distinct fatty alcohols and C7 carbohydrates. The most common health outcome targeted by avocado nutrients is cardiometabolic function, which has broad impacts across multiple organ systems and disease states. However, to our knowledge, few nutrients and compounds were confidently measured with little variability underscoring the need for further characterization of avocado nutrients and potential bioactives. Further characterization would enhance precision nutrition efforts including the development of avocado research to interrogate health benefits in different populations across their lifespans.

## Data Availability

Not applicable.

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
