# Peer review of "Nutritional Composition of Hass Avocado Pulp"

_foods, 2023, doi:10.3390/foods12132516_

Round 1
Reviewer 1 Report
To whom it may concern,
The current review manuscript entitled "Nutritional Composition of Hass Avocado Pulp" is a well-written manuscript. However, it needs minor corrections. My specific comments are shown in the pdf file.
Thank you

Minor corrections are needed. before publication
Author Response
We thank the reviewer for their thoughtful comments.
Attached is our response to your comments. Please also refer to the submitted revised manuscript with tracked changes according to your suggestions. We hope these edits are to your satisfaction.

Reviewer 2 Report
Dear Authors,
The presented manuscript is very well organized and systematically reports the overall nutritional composition and quality of Hass avocado.
- In Table 8, I am finding confusing the first two rows showing total phenolic content, especially with regard to the reported unit mg GAE or mg? How was the phenolic content in the second row determined? If it is projected on fresh weight, explain from what?
- For phenolic acids please correct p and m to italic
- Why are the feruloylquinic acids, quinic acid presented as organic acids in Table 9 and not included in previous table as polyphenolic compounds
Author Response
We appreciate the reviewer for their comments on our manuscript below are our responses, and attached you'll find the revised manuscript with tracked changes. We hope these edits are to your satisfaction.
The presented manuscript is very well organized and systematically reports the overall nutritional composition and quality of Hass avocado.
Thank you for your comment.
- In Table 8, I am finding confusing the first two rows showing total phenolic content, especially with regard to the reported unit mg GAE or mg? How was the phenolic content in the second row determined? If it is projected on fresh weight, explain from what?
We agree with the reviewer. This can be confusing. Total phenolic content was calculated using true reference standards (i.e., not an approximation using an equivalent like gallic acid). We clarified the GAE in total phenol content and the other compounds within the phenolics and organic acids tables that were calculated using equivalent measures to approximate values.
- For phenolic acids please correct p and m to italic
This has been changed in the revised manuscript.
- Why are the feruloylquinic acids, quinic acid presented as organic acids in Table 9 and not included in previous table as polyphenolic compounds
Thank you for your comment. Organic acids and phenolics were organized together at one point during the drafting process and this was overlooked in the editing process. We have revised this in the manuscript.

Reviewer 3 Report
The aim of the article is to collect information previously published or reported on the chemical composition of ripe avocado pulp. In addition to discussing the content of individual components in avocado pulp, an interesting chapter is the one that points to the challenges of proper nutrition. The article is interesting, the results are presented in clear tables and graphs. The text presents the most important conclusions contained in publications on avocado. I recommend that you provide some information, so that it is both interesting from the scientific and practical point of view.
In the introduction, the pro-health properties of Avocados are presented, but no information is provided whether the article that compiles the research results is significant. I think a few sentences about the purpose of this article should be added to the introduction.
Also some information would be helpful, talking about the technological use of avocado. The publication gives the composition of avocado pulp, so it would be important to indicate how such pulp is used in industry and whether the consumer can find avocado in any products.
I understand that the article is about the nutrients contained in ripe avocado pulp. It is stated that it is a climacteric fruit. However, little is known about changes in fruit ripening time after harvesting. I think that the article will be of greater practical importance, especially for consumers, as some information about how to store fruit will be added to ensure that it has the best health-promoting properties.
I propose to change the caption of table number 4 to "Total protein and amino acid content in fresh hass avocado pulp.
The conclusion lacks a summarizing view of the issue under discussion. Is there enough research on the composition of avocado fruit, or do some ingredients require more extensive analytical research? What research should be carried out in the future, what questions we do not know the answers to?. Such a forward-looking view should be written in the summary of the review article.
Author Response
We appreciate the reviewer's thoughtful comments. Our responses to your comments are below. Please also refer to the revised manuscript with tracked changes. We hope these edits are to your satisfaction.
The aim of the article is to collect information previously published or reported on the chemical composition of ripe avocado pulp. In addition to discussing the content of individual components in avocado pulp, an interesting chapter is the one that points to the challenges of proper nutrition. The article is interesting, the results are presented in clear tables and graphs. The text presents the most important conclusions contained in publications on avocado. I recommend that you provide some information, so that it is both interesting from the scientific and practical point of view.
Thank you for your assessment of our manuscript. We have elaborated on your suggestion to include practical information in a comment below.
In the introduction, the pro-health properties of Avocados are presented, but no information is provided whether the article that compiles the research results is significant. I think a few sentences about the purpose of this article should be added to the introduction.
We agree with the reviewer's comment. The last paragraph in the introduction has been revised to better clarify the value of our review.
Also some information would be helpful, talking about the technological use of avocado. The publication gives the composition of avocado pulp, so it would be important to indicate how such pulp is used in industry and whether the consumer can find avocado in any products.
If I understand the reviewer's suggestion, additional information relating avocado to the consumer could be helpful. I think this could be especially helpful for a naive consumer. We have added a sentence in the introduction on foods you can find avocados and the frequently consumed companion foods.
I understand that the article is about the nutrients contained in ripe avocado pulp. It is stated that it is a climacteric fruit. However, little is known about changes in fruit ripening time after harvesting. I think that the article will be of greater practical importance, especially for consumers, as some information about how to store fruit will be added to ensure that it has the best health-promoting properties.
In theory, this is an excellent suggestion. However, in reality, agricultural practices and seasonality seem to play the biggest role in nutrient composition, which is something consumers have little to no control over. WE have added a preparation technique to potentially increase the carotenoid content of the fresh product consumed, given that the carotenoid content is highest in the flesh near the peel. These two comments are included in the revised manuscript discussion.
I propose to change the caption of table number 4 to "Total protein and amino acid content in fresh hass avocado pulp.
This has been changed.
The conclusion lacks a summarizing view of the issue under discussion. Is there enough research on the composition of avocado fruit, or do some ingredients require more extensive analytical research? What research should be carried out in the future, what questions we do not know the answers to?. Such a forward-looking view should be written in the summary of the review article.
We agree with the reviewer's comment. We struggled to draft this section, especially with the breadth of information included in our review. Your outside has provided helpful insight, and we have reworked the conclusion accordingly.
